# Bimetal–Organic Framework-Derived CoMn@C Catalysts for Fischer–Tropsch Synthesis

Linyan Yang, Yu Gao, Yupeng Guo, Zhengjia Li, Jie Cen, Nan Yao and Xiaonian Li *

Institute of Industrial Catalysis, College of Chemical Engineering, Zhejiang University of Technology, Hangzhou 310014, China; yanglinyan2007@126.com (L.Y.); gaoyu899@163.com (Y.G.); guoyupeng997@163.com (Y.G.); lizhengjia@zjut.edu.cn (Z.L.); jcen@zjut.edu.cn (J.C.); kenyao@zjut.edu.cn (N.Y.)
* Correspondence: xnli@zjut.edu.cn

**Abstract:** Introducing promoters to cobalt-based catalysts for Fischer–Tropsch synthesis (FTS) has been found to be efficient in adjusting their performance in converting syngas into long-chain hydrocarbons. High spatial proximity of the promoter and reactive metal is desired to maximize the effectiveness of the promoters. In this work, CoMn@C composites were synthesized by the one-step carbonization of bimetal–organic frameworks (BMOFs: CoMn-BTC). BMOF-derived catalysts naturally exhibited that cobalt nanoparticles (NPs) are confined in the carbon matrix, with a concentrated particle size distribution around 13.0 nm and configurated with MnO highly dispersed throughout the catalyst particles. Mn species preferentially bind to the surfaces of Co NPs rather than embedded into the Co lattice. The number of Mn–Co interfaces on the catalyst surface results in the weakened adsorption of H but enhanced adsorption strength of CO and C. Hence, the incorporation of Mn significantly inhibits the production of $CH_4$ and $C_2$–$C_4$ paraffin boosts light olefin ($C_{2-4}^{=}$) and $C_5^{+}$ production. Furthermore, the FTS activity observed for the Mn-promoted catalysts increases with increased Mn loading and peaks at 2Co1Mn@C due to the abundance of Co–Mn interfaces. These prominent FTS catalytic properties highlight the concept of synthesizing BMOF-derived mixed metal oxides with close contact between promoters and reactive metals.

**Keywords:** Fischer–Tropsch synthesis; bimetallic–organic frameworks; CoMn@C composites; Mn–Co interfaces

## 1. Introduction

As an important route to convert coal, biomass, and/or carbon-containing wastes into ultra-clean liquid hydrocarbon fuels via synthesis gas (syngas, CO+$H_2$), Fischer–Tropsch synthesis (FTS) has received renewed attention due to the gradually matured gasification technology, increased concern about energy, and the implementation of more stringent environmental legislation on liquid fuel [1–3]. Co-based catalysts have characteristics such as a reasonable cost, high activity, superior selectivity, and low $CO_2$ emissions [4,5]. However, without modification with a structural or electronic promoter, cobalt-based FTS catalysts always produce large amounts of methane. Introducing promoters to cobalt-based catalysts for Fischer–Tropsch synthesis has been found to be efficient in adjusting their performance in converting syngas into long-chain hydrocarbon and even promoting the catalytic activity [6–9]. Mn is a promoter of particular interest for Fischer–Tropsch synthesis catalysts, which could increase the catalyst activity, decrease $CH_4$ formation, increase the olefin to paraffin (O/P) ratio of the $C_2$–$C_4$ fraction, and enhance the formation of $C_5^{+}$ products [10–13]. A number of studies state that Mn in close proximity to Co is essential for the manifestation of Mn's promotion effects [14–16]. A connection can be made between catalyst performance and the spatial association between Co and Mn [13]. Therefore, developing efficient catalyst preparation methods capable of ensuring this close contact between the promoter and metal is of particular importance in promoting the FTS catalytic performance.

In fact, the thermal decomposition of metal complexes has long been used as a method to regulate the elemental distribution and size of metal particles over supported catalysts [17]. Recently, metal–organic frameworks (MOFs) have emerged as promising precursors for the synthesis of nanomaterials. MOF-derived nanoporous carbon materials have attracted significant interest due to their advantages of controllable porosity, good thermal/chemical stability, and easy modification with other elements and materials [18–21]. More recently, Basolite F300, ZIF-67, and Co-MOF-74 were used to prepare a catalyst for FTS [22–24]. Compared with traditional supported catalysts, MOF-derived catalysts possess very a high cobalt loading, while they contain well-dispersed active metal Co nanoparticles embedded in the carbon matrix [24]. The confinement and high dispersion result in high activity and enhanced stability. To maximize the synergistic effect between the promoter and reactive metal, bimetal–organic frameworks (BMOFs) are widely chosen as self-sacrificial templates to prepare bimetallic composites, basically via thermolysis [25–27]. Two metal nodes are co-complexed with organic ligands in the same MOF, and the uniform distribution of different metal oxide species would result in their strong interaction [28]. BMOFs of Ni-Co-BTC solid microspheres with diverse Ni/Co molar ratios are readily prepared by solvothermal methods to induce the $Ni_xCo_{3-x}O_4$ composites as anode materials [29]. Moreover, Fe-Co based catalysts derived from a Co-Fe-MOF exhibit high conversion activity for renewable biomass resources [30]. This strategy does seem to be possible for the synthesis of bimetallic MOFs, where these two metals have close ionic radii and electronegativity. To date, there are no reports on bimetallic MOF-derived Co-based FTS catalysts that have intimate contact between Co and the introduced promoters.

Thus, in this work, a series of CoMn-BTC with different Co/Mn molar ratios were chosen as the bimetallic MOF precursors to synthesize a Co-based Fischer–Tropsch synthesis catalyst with close contact between the active metal Co and promoter Mn. Mn species preferentially bind to the surfaces of Co NPs, rather than being embedded into the Co lattice. Due to the abundant Co–Mn interfaces, the resulting CoMn@C catalysts display outstanding FTS activity, along with lower $CH_4$ selectivity and enhanced olefin ($C_2$–$C_4^=$) and $C_5^+$ selectivity.

## 2. Results and Discussion

### 2.1. FTS Catalytic Performance

To fabricate the FTS catalysts, we used the as-synthesized MOFs as self-sacrificial templates and thermally treated them to carbonize the organic components as well as reduce the metal species. After treatment, the as-synthesized composites Co@C and CoMn@C were used as catalysts in the FTS. The FTS catalytic performance of the MOF-derived catalysts with different Mn/Co molar ratios (Mn/Co molar ratios of 0, 0.05, 0.1, 0.5, and 1.0) was evaluated. We compared the FTS performance of these catalysts through limiting the CO conversion to ~20% by appropriately adjusting the space velocity employed in the reactions. The steady-state CO conversions and product selectivities for the catalysts are presented in Figure S1. It could be seen that all the MOF-derived catalysts could maintain good stability against deactivation when running the reaction over 25 h. Moreover, it was visible that the deactivation rate of the catalysts was weakened when increasing the Mn loading, and 1Co1Mn@C presented the best stability. Compared with the traditional carbon-supported 2Co1Mn/AC catalyst, BMOF-derived 2Co1Mn@C presented better stability, higher $C_5^+$ selectivity, and lower $CH_4$ selectivity. Furthermore, it can also be observed in Figure S1 that there was more rapid deactivation during the initial stages of the reaction, and thus the activities and product selectivities were acquired in the time on stream (TOS) of 20–25 h, in which section the catalysts were stable. The catalytic activities were normalized by the Co weight with the CO consumption. As shown in Figure 1a and Table S1, with an increase in the Mn/Co molar ratio from 0 to 1.0, the content of Mn increases to a maximum of 15.4 wt.% accompanied by a decrease in Co content from 57.4 wt.% to 33.0 wt.%, as determined by the ICP-MS result. Surprisingly, the 20Co1Mn@C catalyst with an ultra-low Mn concentration (1.7 wt.%) exhibited an almost two-fold higher cobalt time yield (CTY)

value ($1.20 \times 10^{-5}$ $mol_{CO} \cdot g_{Co}^{-1} \cdot s^{-1}$) compared to Co@C ($0.68 \times 10^{-5}$ $mol_{CO} \cdot g_{Co}^{-1} \cdot s^{-1}$), indicating the positive contribution of Mn additions. With a further increase in the Mn/Co molar ratio to 0.1 and 0.5, the CTY increased slightly to $1.25 \times 10^{-5}$ $mol_{CO} \cdot g_{Co}^{-1} \cdot s^{-1}$ over 10Co1Mn@C and $1.60 \times 10^{-5}$ $mol_{CO} \cdot g_{Co}^{-1} \cdot s^{-1}$ over the 2Co1Mn@C sample, and subsequently decreased to $0.55 \times 10^{-5}$ $mol_{CO} \cdot g_{Co}^{-1} \cdot s^{-1}$ over the 1Co1Mn@C sample.

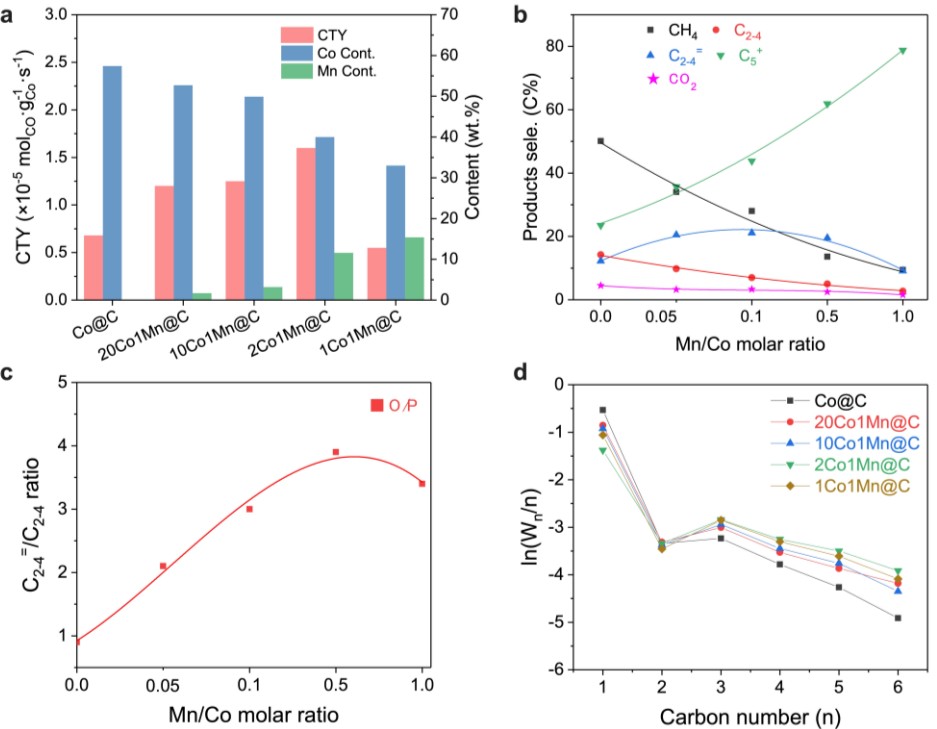

**Figure 1.** (**a**) Cobalt time yield (CTY) and content of Co, Mn measured by ICP-MS; (**b**) product selectivity; (**c**) $C_2$–$C_4$ olefin/paraffin ratio (O/P) as a function of Mn/Co molar ratio, and (**d**) ln(Wn/n) obtained by the catalysts at 235 °C under 1 bar with CO conversion limited to ~20% by adjusting the space velocity. Here, the GHSV = 5000 $mL \cdot g_{cat}^{-1} \cdot h^{-1}$ for Co@C, GHSV = 8000 $mL \cdot g_{cat}^{-1} \cdot h^{-1}$ for 20Co1Mn@C, 10Co1Mn@C, and 2Co1Mn@C, GHSV = 2400 $mL \cdot g_{cat}^{-1} \cdot h^{-1}$ for 1Co1Mn@C.

The product selectivity of $CH_4$, $C_2$–$C_4$, and $C_5^+$, as well as the $C_2$–$C_4$ olefin/paraffin ratio (O/P), is presented in Figure 1b,c. With an increase in the Mn/Co molar ratio from 0.0 (Co@C) to 1.0 (1Co1Mn@C), the incorporation of Mn significantly inhibits the production of $CH_4$, $CO_2$, and $C_2$–$C_4$ paraffin accompanied by an enhancement in the $C_5^+$ selectivity. Accordingly, methane selectivity decreases from 50.1% to 9.4%, and selectivity towards long $C_5$+ ranges from 23.5% over Co@C to 78.8% over 1Co1Mn@C. Meanwhile, the $C_2$–$C_4$ olefin ($C_{2-4}^=$) selectivity first increases and reaches a selectivity plateau along with Mn addition. The $C_{2-4}^=$ selectivity on 10Co1Mn@C (21.1%) and 2Co1Mn@C (19.5%) is almost two-fold higher than that of unpromoted Co@C catalysts (12.2%). A further increase in the Mn loading to 15.4 wt.% (1Co1Mn@C) gives rise to a decrease in $C_{2-4}^=$ selectivity, and thus the olefin/paraffin ratio (O/P) value increases and peaks in the 2Co1Mn@C sample. Moreover, the product distribution (Figure 1d) shifts to longer hydrocarbons with the increase in Mn loading, implying an increase in the chain growth probability ($\alpha$). The intrinsic activity is usually affected by the nature of the catalyst's surface structure and the electronic interaction between the catalytic surface and the supports or additives. We thus examined the active surface areas, electronic interactions with additives, and surface structures of these catalysts.

## 2.2. Morphology and Crystal Structure of Co-BTC and 2Co1Mn-BTC Precursors

Co-BTC and 2Co1Mn-BTC, as self-sacrificing templates, were synthesized according to the literature procedure [31]. As shown by the scanning electron microscope (SEM) images (Figure S1), Co-BTC and 2Co1Mn-BTC formed a 3D prism structure that consists of agglomerated sheets. The crystal structure of Co-BTC and 2Co1Mn-BTC was confirmed by XRD, as shown in Figure 2a. Co-BTC and 2Co1Mn-BTC have a monoclinic crystal structure [32]. The evident similarity of the XRD patterns of the 2Co1Mn-BTC bimetallic MOF and Co-BTC indicated that they are isostructural crystals. Notably, the diffraction peaks of the 2Co1Mn-BTC sample shifted left obviously. This phenomenon is consistent with the fact that the ionic radius of $Mn^{2+}$ is larger than that of $Co^{2+}$.

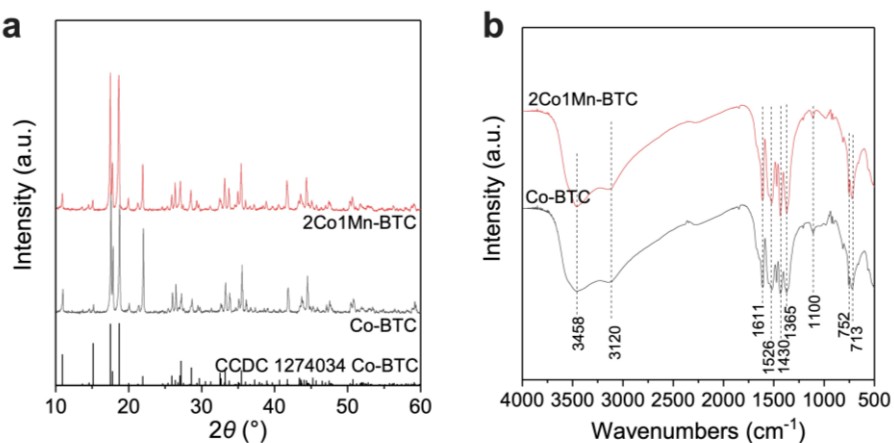

**Figure 2.** (**a**) XRD patterns and (**b**) IR spectrum of Co-BTC and 2Co1Mn-BTC precursors.

Figure 2b presents the IR spectrum of Co-BTC and 2Co1Mn-BTC. The peaks in the range of 700–900 cm$^{-1}$ correspond to the C-H wagging vibration of benzene. The band at 3458 cm$^{-1}$ and 3120 cm$^{-1}$ implies the presence of coordinated/crystal water and an O-H⋯O bond in the Co-MOF, respectively. There is no absorption peak in the range of 1720–1680 cm$^{-1}$, indicating that the carboxyl group on the H$_3$BTC has been completely deprotonated. The peaks observed at 1361, 1430, 1526, and 1611 cm$^{-1}$ result from the symmetric (Vs) and asymmetric (Vas) stretching vibrations of carboxylate groups (-COO-). Moreover, the splitting of the asymmetric vibration band reveals that the carboxylate groups have two different coordination modes [33]. The IR spectrum of 2Co1Mn-BTC is consistent with that of Co-BTC, implying the same coordination structure among them.

## 2.3. Mn Promotion Effect on the Catalysts

From the N$_2$ sorption measurements at −196 °C, it was observed that the Co@C samples exhibited a Brunauer–Emmett–Taller (BET) surface area of 169.7 m$^2$/g. The MOF-derived FTS catalysts that we have prepared here come from the direct pyrolysis of the MOF precursors and thus contain a very high weight percentage of cobalt in the final catalysts (57.36 wt.% for Co@C). The high cobalt content in the MOF-derived FTS catalysts contributes to the observed modest surface areas. Moreover, the surface areas of the BMOF-derived CoMn@C composites were related to the Mn/Co molar ratio, as listed in Table S2; the surface area decreased from 148.0 m$^2$/g to 125.5 m$^2$/g when increasing the Mn content from 3.2 wt.% (10Co1Mn@C) to 11.6 wt.% (2Co1Mn@C). As shown in Figure 3a–c, all the Co@C and CoMn@C composites derived from Co-MOF and CoMn-MOF exhibited a type I(b) isotherm with a H4 type hysteresis loop according to the IUPAC classification. The high uptake at low pressure and small hysteresis loop indicate that there is a large number of micropores in the MOF-derived catalysts' particles. From the pore size distribution (illustration), it was observed that the MOF-derived Co@C, 10Co1Mn@C, and 2Co1Mn@C presented similar micropore distributions. Therefore, the rapid deactivation during the initial stages of the reaction (Figure S1) could be related to the diffusional limitations

caused by the initially smaller pores within the MOF-derived catalysts, which were filled by the liquid waxes generated under the reaction conditions. Moreover, the introduction of Mn narrowed the pore distribution and significantly reduced the number of pores with a diameter of less than 1 nm. We speculate that the reason for this is that the formation of MnO species restrains the shrinkage and collapse of the pore in the carbon matrix during the pyrolysis process.

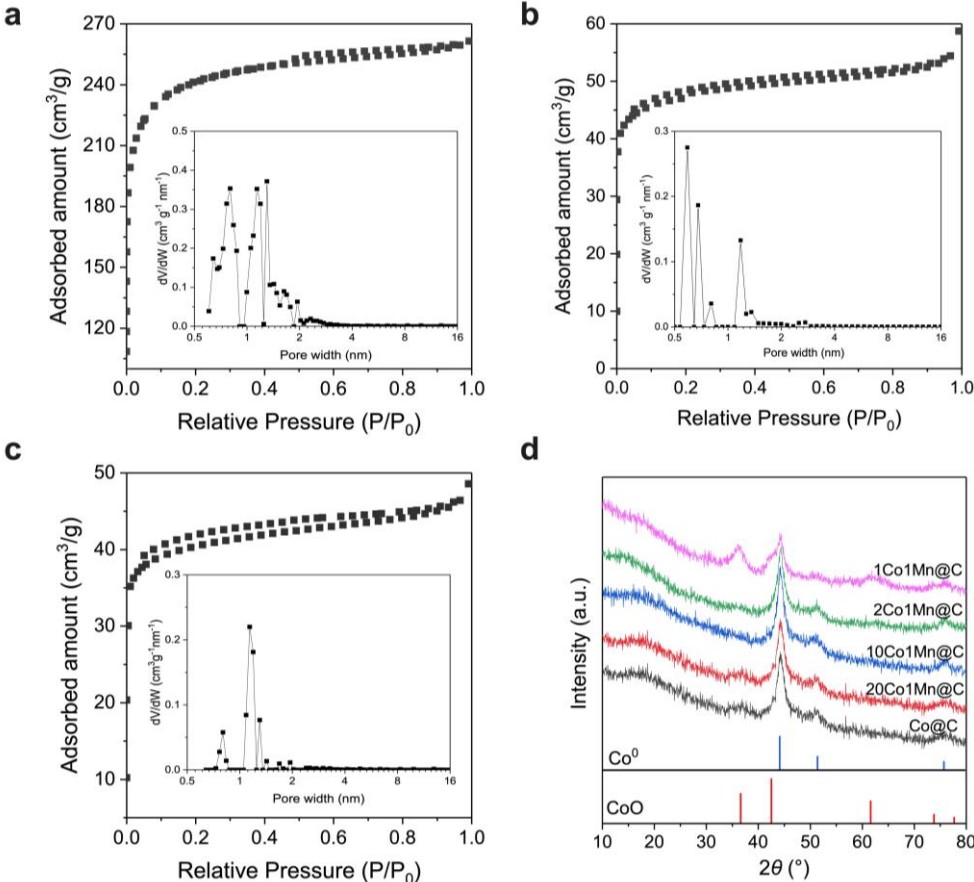

**Figure 3.** $N_2$ adsorption–desorption isotherms and pore size distribution: (**a**) Co@C, (**b**) 10Co1Mn@C, (**c**) 2Co1Mn@C. (**d**) XRD patterns for MOF-derived CoMn@C composites.

The crystal structures of the as-synthesized Co@C and CoMn@C composites were evaluated by XRD, as illustrated in Figure 3d. The diffraction peaks (44.3°, 51.3°, and 76.1°) observed for MOF-derived composites can be assigned to metallic cobalt with a face-centered cubic (fcc) phase. Except for the 1Co1Mn@C sample, the diffraction peaks (36.5°, 42.4°, 61.5°, 73.7°) of the CoO phase were detected, implying that the superfluous loading of the Mn promoter inhibited the reduction of CoO to $Co^0$. Compared to Co@C, the shifts in the $Co^0$ diffraction peaks were undetected, indicating that the CoMn composite oxide was not present in these BMOF-derived CoMn@C samples, even with a maximum Mn loading of 15.4 wt.% (1Co1Mn@C). Furthermore, no other detectable diffraction peaks assigned to Mn species appeared in the XRD patterns, implying that Mn was highly dispersed or existed as an amorphous species in this series of CoMn-MOF-derived catalysts. Figure S3 shows the XRD patterns for 2Co1Mn@C before and after reduction. The crystallite sizes for 2Co1Mn@C and reduced 2Co1Mn@C are 7.3 nm and 7.7 nm, respectively. Therefore, it is worth noting that the Co crystallite size has no obvious increase after reduction at 350 °C for 6 h under 10%$H_2$/Ar. Furthermore, the amorphous cobalt oxides formed by passivation could be almost completely reduced to metal cobalt. As shown in Figure S4, the peak at around 250 °C can be assigned to the reduction of cobalt oxides on the surface, and no hydrogen consumption peak could be observed above 350 °C. As presented in

Table S2, the addition of Mn slightly reduces the grain sizes of $Co^0$. The crystallite sizes of the $Co^0$ over the MOF-derived catalyst are approximately equal to 7.0–9.0 nm. The surface density of metal cobalt (8.9 g·mL$^{-1}$, 15.1 Co atoms nm$^{-2}$) was applied in the TOF calculation, as illustrated in Table S2. Without the particle size effect of the Co nanoparticles, the TOF value increased from $3.48 \times 10^{-3}$ s$^{-1}$ (Co@C) to $6.91 \times 10^{-3}$ s$^{-1}$ (2Co1Mn@C) when increasing the Mn content to 11.6 wt.%, indicating that the Mn promoters affected the electronic structure of the Co species and thus improved the catalytic activity.

The morphology of the MOF-derived composites was characterized by scanning electron microscopy (SEM) and transmission electron microscopy (TEM), as shown in Figure 4. Even after carbonization, the obtained catalysts inherited the original morphology of the MOF precursor, although the particle surface was distorted (Figure 4a,d,g). The TEM images of Co@C (Figure 4b), 20Co1Mn@C (Figure 4e), and 2Co1Mn@C (Figure 4h) indicate that these composites are composed of many uniformly distributed nanoparticles, which are confined within the porous carbon framework, regardless of the different Mn/Co molar ratios. A high-resolution TEM image confirmed the existence of crystallized Co nanoparticles. Lattice fringes were coherently extended on each nanoparticle, indicating that each particle had a single crystalline structure (Figure 4i). The lattice fringes with the interplanar spacing of approximately 2.08 Å were determined to be associated with the (111) facet of the fcc $Co^0$ crystal. Based on these analyses, all the mean particle sizes of Co nanoparticles were around 13.0 nm for the catalysts with different Mn loadings. The particle sizes measured by TEM were higher compared to those estimated by XRD (approximately 7–9 nm). In the process of MOF pyrolysis in an inert atmosphere, the organic ligands could be decomposed into reducing substances such as CO, $H_2$, and C to reduce $Co^{2+}$ to metallic cobalt. In order to prevent the catalyst from excessive oxidation when exposed to air, the MOF-derived composites were pre-passivated at room temperature in $1\%O_2/N_2$, so an amorphous oxide layer with a thickness of around 2 nm formed on the surfaces of the metal cobalt particles. The amorphous oxide on the surface could not be detected by XRD, and thus the crystallite sizes of the metal Co calculated based on the Scherer equation were slightly smaller than the particle sizes measured by TEM. The possibility that the observed selectivity differences for the MOF-derived catalysts with different Mn/Co molar ratios were due to nanoparticle size effects is unlikely given that the mean diameters of the catalysts were similar. The direct carbonization of Co dispersed in different polymers usually results in the formation of cobalt nanoparticles with a very broad particle size distribution [17]. Meanwhile, in the case of Co-MOF-derived composites, the particle size distribution of Co crystallite is narrower.

The optimal combination was high dispersion with the maximum loading, anchored firmly in a stable matrix. The morphology and elemental distribution of the BMOF-derived CoMn@C composites were further examined by high-angle annular dark-field (HAADF) scanning transmission electron microscopy (STEM) imaging (Figure 4c,f,j) and energy-dispersive X-ray spectroscopy (EDS) mapping. The MnO and Co particles are significantly brighter than the MOF-derived carbon matrix since the HAADF image intensity is proportional to the atomic number [34]. It can be seen that the C and O elements are homogeneously distributed. The Co element mapping shows that nanoparticles mainly consist of Co atoms and all Co atoms are concentrated on the embedded nanoparticles. However, elemental mapping of the 20Co1Mn@C sample showed an even distribution of Mn. Mn species on the Co particle are highly dispersed without aggregation; some Mn atoms are also detected on other sections where no Co elements are observed. A reasonable spatial association between Co and Mn is that small Mn nanoparticles decorate the surfaces of larger Co nanoparticles. The above analysis concludes that the highly dispersed MnO could yield plenty of Mn–Co interfaces. Further increasing the Mn/Co ratio to 0.5 (2Co1Mn@C), the Mn species dispersed on cobalt particles become more abundant compared to the previous catalyst 20Co1Mn@C, while there is still no sign of MnO aggregation on the Co particles, highlighting the importance of the framework structure of the precursor in the yield of dispersed Mn species for promotion. A reasonable interpretation is that the

confinement structure restrains the migration and agglomeration. It is worth noting that, although the Mn is in close association with Co, this preparation method seems to limit the formation of Co–Mn bulk solid solutions, even with MnO loading close to 11.6 wt.%, as seen previously in the abovementioned XRD characterization. When the Mn additive exceeds a certain amount, the distribution on the Co surface would evolve from isolated dispersion into a continuous layer, which explains why the reaction activity of 1Co1Mn@C decreased obviously compared with 2Co1Mn@C due to surface blockage.

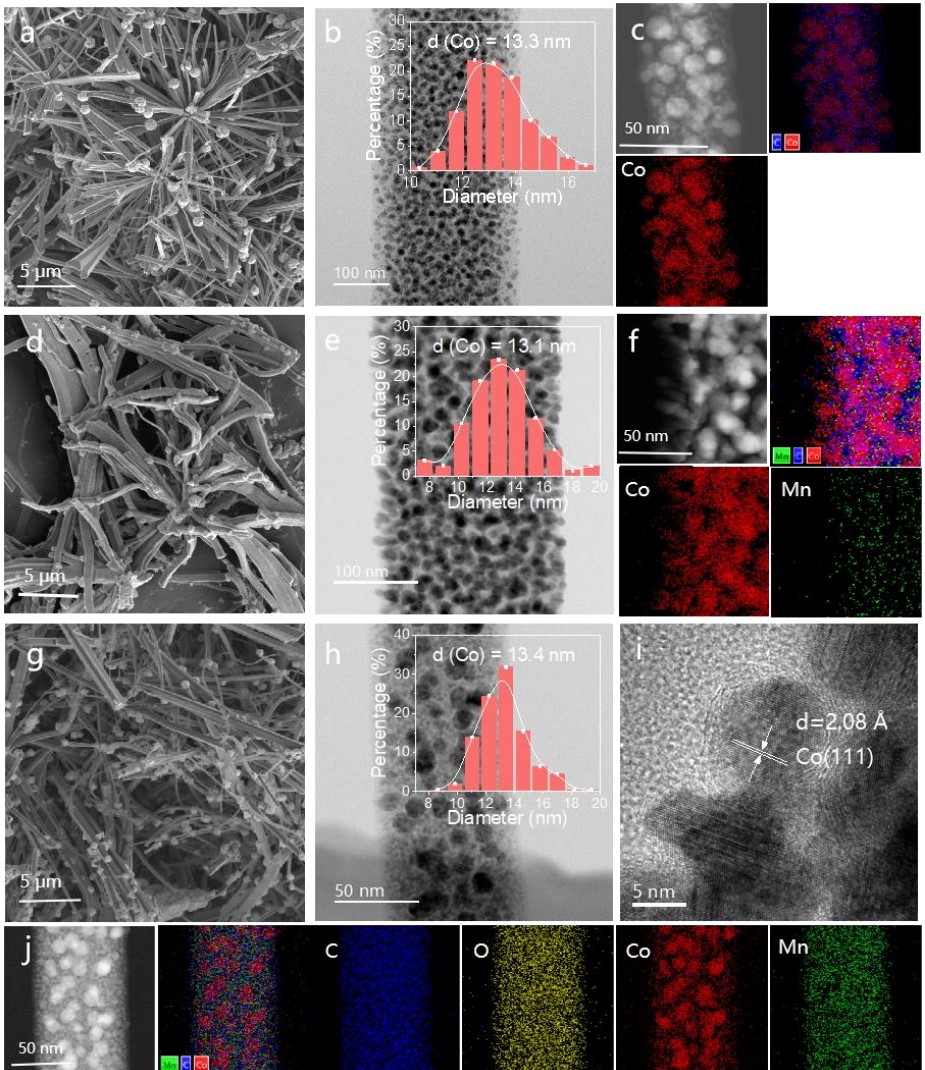

**Figure 4.** Scanning electron microscopy (SEM) images: (**a**) Co@C, (**d**) 20Co1Mn@C, (**g**) 2Co1Mn@C. Transmission electron microscopy (TEM) images: (**b**) Co@C, (**e**) 20Co1Mn@C, (**h**) 2Co1Mn@C. Insets in (**b**,**e**,**h**) show the Co$^0$ particle size distribution. (**i**) High-resolution transmission electron microscopy (HRTEM) image for 2Co1Mn@C. (**c**,**f**,**j**) HAADF-STEM images and corresponding EDS mapping images of Co and MnO nanocrystals homogeneously embedded in porous carbon matrix.

XPS characterization was conducted to analyze the surface elemental composition as well as the chemical bonding environment of the as-synthesized catalysts, and all binding energy values were calibrated to C 1 s at 284.8 eV. As presented in Figure 5a, the surface chemical composition of the MOF-derived catalysts is mainly C species, suggesting a carbon encapsulation structure. Moreover, as shown in Table S1, the surface Mn/Co molar ratios for CoMn@C composites are much higher than those determined experimentally by ICP-MS, implying that the enrichment of Mn occurred on the catalyst surface. Figure 5b–d show the XPS spectra of Co 2p, Mn 2p, and O 1s. The high-resolution Co 2p XPS spectrum

(Figure 5b) of Co@C exhibits four fitting peaks of $Co^0$ (777.8 eV), $Co^{3+}$ (779.5 eV), and $Co^{2+}$ (782.1 eV) along with a satellite peak (786.0 eV). As we know, $Co^0$ species can be oxidized to $Co^{3+}$ or $Co^{2+}$ when passivated in a $1\%O_2/N_2$ atmosphere, thus forming a layer of oxide film on the surfaces of metal cobalt particles. The content ratio of $Co^0/(Co^{3+} + Co^{2+})$ changes from 28.12% (Co@C) to 24.77% (2Co1Mn@C) with increasing Mn loading to 11.6 wt.%. Further increasing the Mn loading to 15.4 wt.%, the peak at 777.8 eV assigned to $Co^0$ cannot be detected over 1Co1Mn@C, implying that the presence of excess Mn species inhibits the reduction of $Co^{2+}$ to $Co^0$ during the pyrolysis process, which is consistent with the XRD results (Figure 3c). With the introduction of Mn, the binding energies of $Co^{2+}$ and $Co^{3+}$ shift to lower values, while no obvious shift in the $Co^0$ binding energy can be observed, suggesting that despite the close contact of Mn and the passivation oxide layer on the surface, the bulk $Co^0$ remains stable. It is reasonable to infer that Mn preferentially forms isolated nanoparticles rather than being embedded in the Co lattice. Furthermore, the peak of Mn 2p3/2 is located at the binding energy of 641.2 eV (Figure 5c); the presence of a satellite feature at 647 eV and the absence of an identifiable shape at the top of the $Mn2p_{3/2}$ indicate that the possible oxidation state of Mn species is +2. This result is in good agreement with previous publications on MOF-derived Fe3Mn1/N-CNTs-100 [35]. Additionally, the O 1s spectrum (Figure 5d) can be decomposed into three independent feature peaks. The peak at 533.2 eV is assigned to the small amounts of physically and chemically adsorbed water molecules at the surface region [36]. The peak at 531.6 eV can be ascribed to the surface chemisorbed oxygen species, while the peak at 530.1 eV is ascribed to the typical metal–oxygen bond (M-O). The increase in MnO species results in an incremental M-O ratio.

### 2.4. Theoretical Investigations

To understand the activity and product selectivity of FTS on MOF-derived composites' catalytic surfaces, DFT calculations were performed to evaluate the promotion effect of Mn in catalysis and the possible position of Mn. As the XPS results revealed the enrichment of Mn on the catalyst's surface, three Mn-modified models were chosen with Mn at the surface (Mn-on-Co) and subsurface (Mn-in-Co), as well as embedded in the Co lattice (Mn-under-Co). For simplicity, Mn species were modeled as $Mn^0$ rather than MnO for our DFT calculations [8,37]. Therefore, the adsorption binding energies of H, C, and CO species were calculated on the surfaces of Co(111), Mn-on-Co(111), Mn-in-Co(111), and Mn-under-Co(111), respectively. As shown in Figure 6, the calculated adsorption energies on clean Co(111) are in reasonable agreement with the values reported in the literature [8,37]. Interestingly, Mn-on-Co(111), Mn-in-Co(111), and Mn-under-Co(111) all lead to higher binding energies for CO. Meanwhile, the effect is stronger for Mn-on-Co(111) compared with the clean Co(111) surface, with changes in the adsorption energies being in excess of 1 eV, but it is still significant on the Mn-under-Co surface without direct Mn–adsorbate contact.

To analyze the charge distribution between Co and Mn atoms, the charge density difference was calculated. In the illustration in Figure S5, the cyan and yellow colors represent the depletion and accumulation of charge. This charge transfer can be easily explained on the basis of the electronegativity of transition metals (Co (1.88) and Mn (1.55)). It can be clearly seen that the introduction of Mn leads to an electron charge gain at the Co surface, which results in increased back-donation into the antibonding π* orbital of CO and thus weakens the C–O bond. Therefore, compared with the clean Co(111) surface, the C–O bond cleavage on the Mn-modified Co(111) surfaces is more energetically favorable. Consequently, the intrinsic activity observed for the Mn-promoted catalysts increases with increasing Mn loading and peaks for the 2Co1Mn@C sample due to the abundant Co–MnO interfaces. A similar observation has previously been made on Mn-modified Rh [38], where it was observed that the addition of Mn increased the binding energies of CO and decreased the dissociation barrier.

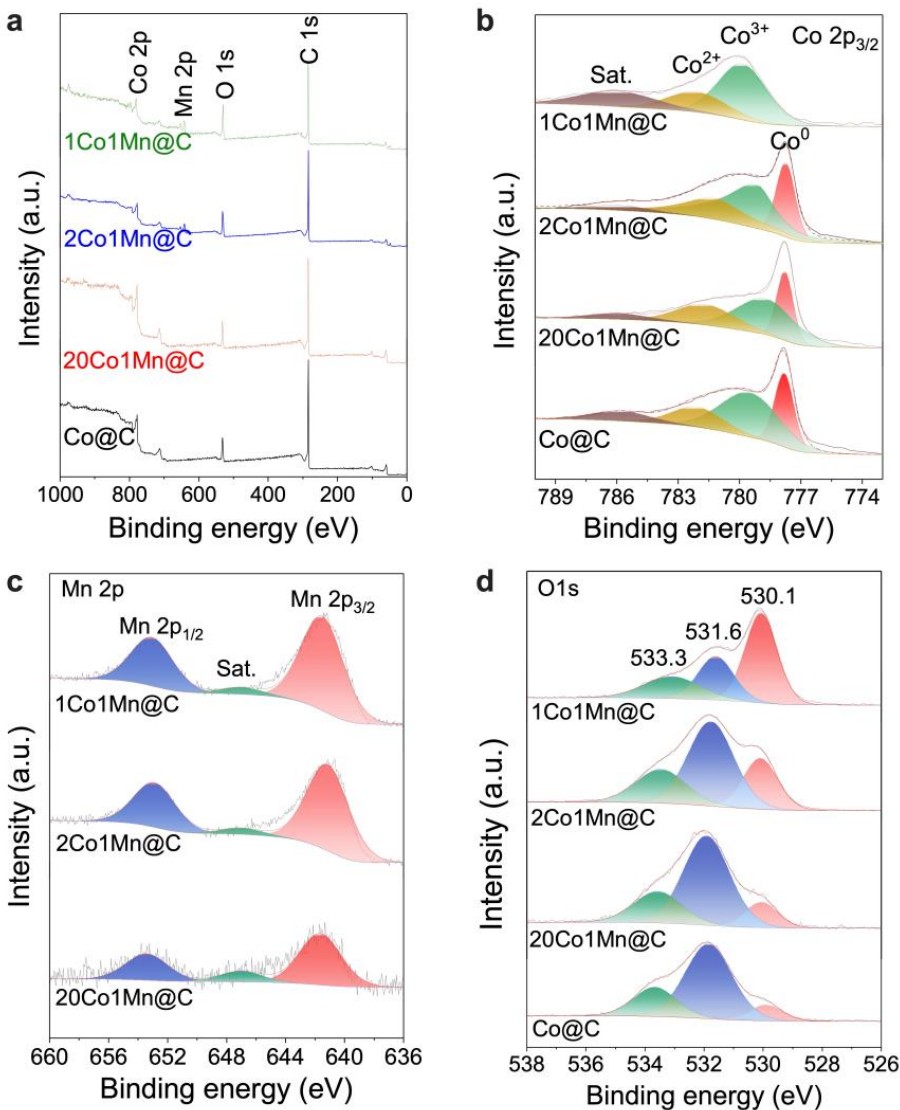

**Figure 5.** (**a**) High-resolution spectra, (**b**) Co 2p, (**c**) Mn 2p, and (**d**) O 1s XPS spectra of Co@C and BMOF-derived CoMn@C catalysts.

Moreover, the adsorption energies for C and H are also strongly correlated with the spatial association between Co and Mn. The presence of Mn clearly enhanced the adsorption energy of C. The Mn-under-Co surface has the lowest C binding energy (−7.86 eV) among the Mn-promoted surfaces, and the Mn-on-Co surface presents the highest adsorption energy (−9.90 eV) for C. However, the Mn-under-Co surface leads to a higher adsorption energy of H (−1.98 eV) than that of Co(111), while the H adsorption energy for the Mn-in-Co surface (−1.39 eV) is approximately equal to that of Co(111) (−1.50 eV). In contrast, Mn-on-Co leads to much lower adsorption energy for H (−0.54 eV). These results indicate that the Mn-on-Co surface has the optimal active sites for C–O bond breaking and C–C coupling to increase the FTS activity. Furthermore, in the case of the Mn-on-Co structure, the much lower adsorption and decreased coverage of H could also inhibit chain termination by hydrogenation, as well as the secondary hydrogenation of olefin, resulting in lower $CH_4$ selectivity and enhanced olefin and $C_5^+$ selectivity, consistent with the experimental results (Figure 1). Therefore, in the case of CoMn-MOF-derived CoMn@C catalysts, it is reasonable to speculate that the introduced Mn species preferentially bind to the surfaces of Co NPs rather than being embedded into the Co lattice. This is also in excellent agreement with the XRD, EDS mapping, and XPS results suggesting that Mn preferentially forms isolated

nanometer particles rather than being embedded in the Co lattice or inducing the formation of mixed Co–Mn oxides.

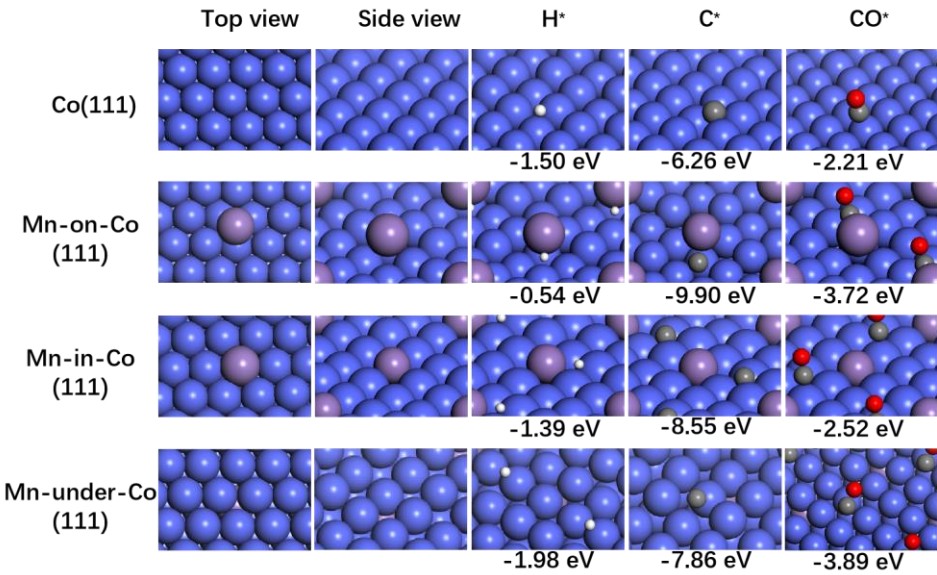

**Figure 6.** DFT-optimized configurations of H, C, and CO binding on Co(111), Mn-on-Co(111), Mn-in-Co(111), and Mn-under-Co(111) surfaces. Blue, purple, grey, red, and white spheres represent Co, Mn, C, O, and H atoms, respectively. "*" represents an adsorption state.

## 3. Materials and Methods

### 3.1. Catalyst Preparation

Chemicals and materials: Cobalt acetate tetrahydrate ($Co(CO_3)_2 \cdot 4H_2O$, 99.5%), manganese acetate tetrahydrate ($Mn(CO_3)_2 \cdot 4H_2O$, 99.0%), and trimesic acid ($H_3BTC$, 98.0%) were purchased from Sigma-Aldrich (St. Louis, MO, USA). All the materials were used as received, without further purification.

Synthesis of Co@C and CoMn@C catalysts: Co-BTC and CoMn-BTC with different Mn/Co molar ratios (Mn/Co = 0.05, 0.1, 0.5 and 1.0) were synthesized following a modified method [31]. Co@C and CoMn@C catalysts were obtained after thermal decomposition of the corresponding Co-BTC and CoMn-BTC precursors. Typically, $Co(CO_3)_2 \cdot 4H_2O$ (1.0 mM) and $Mn(CO_3)_2 \cdot 4H_2O$ (1.0 mM) were dissolved into a mixed solution of ethanol (50 mL) and deionized water (50 mL), forming solution A. $H_3BTC$ (0.90 g) was dissolved into a mixed solution of ethanol (50 mL) and deionized water (50 mL), forming solution B. Then, solution B was poured into solution A under magnetic stirring. The obtained pink precipitates were rinsed with deionized water and ethanol several times, dried at 80 °C for 12 h, and thermally treated in $N_2$ at 500 °C for 4 h, followed by passivation at 30 °C with 1%$O_2$/Ar for 2 h. The obtained samples were designated as 20Co1Mn@C, 10Co1Mn@C, 2Co1Mn@C, and 1Co1Mn@C based on the Mn/Co molar ratio.

### 3.2. Catalyst Evaluation

The FTS measurements at 1 bar were carried out at 235 °C in a tubular quartz fixed-bed reactor (diameter, 8 mm) using $H_2$/CO = 2/1 (*v*/*v*) after reduction treatment at 350 °C for 6 h in 10%$H_2$/Ar flow (8000 mL·$g_{cat}^{-1}$·$h^{-1}$) at atmospheric pressure. Typically, 0.1 g of catalyst particles was diluted with 0.3 g of SiC particles to remove any temperature gradient within the catalyst bed. The catalyst was reduced prior to the reaction at 350 °C with a mixture of $H_2$ and Ar (10%$H_2$ *v*/*v*, 8000 mL·$g_{cat}^{-1}$·$h^{-1}$) for 6 h at atmospheric pressure, and the heating ramp was 2 °C/min. The temperature was then dropped to 180 °C in Ar (99.999%) flow for 30 min to purge the residual reducing gas. Subsequently, the feed flow was switched to a mixture of $N_2$, $H_2$, and CO ($N_2$/$H_2$/CO = 3/64.6/32.3 *v*/*v*/*v*, 3000 mL·$g_{cat}^{-1}$·$h^{-1}$), and the temperature was increased at a 1 °C/min heating rate to 235 °C. During the reaction,

the reactor effluent was analyzed on-line by an Agilent 6890N gas chromatograph (GC) equipped with two columns and two detectors. Analysis of $N_2$ (reference), $H_2$, CO, $CH_4$, and $CO_2$ was performed using a carbon molecular sieve column (TDX-1) and a thermal conductivity detector (TCD), using helium as the carrier gas. Hydrocarbons from $C_1$ to $C_6$ were analyzed using a capillary column (Agilent 19091P-S15HP-AL/S) and a flame ionization detector (FID). The selectivity was calculated as the percentage of equivalent carbons present in the hydrocarbon product (C%). As the space time yields of $C_5^+$ were relatively low at 1 bar, it was difficult to perform a full analysis of all the products. In this case, the method by difference was used to calculate the $C_5^+$ selectivity (CO converted into products other than $CO_2$ and the $C_1$–$C_4$ products), where $C_5^+$ represents hydrocarbons with five or more carbons.

### 3.3. Catalyst Characterization

The infrared spectra were collected using the Nicolet 6700 FT-IR instrument (Thermo Scientific, Waltham, MA, USA) equipped with a low-temperature Mercury–Cadmium–Telluride (MCT) detector. All spectra were obtained with a resolution of 4 cm$^{-1}$ and an accumulation of 64 scans. ICP-MS analyses (Co and Mn) were performed on an Agilent ICP-MS 7700 instrument (Tokyo, Japan). The specific surface area, pore volume, and average pore size of the samples were evaluated using $N_2$ physisorption at $-196$ °C on a Micromeritics 2420 instrument (Norcross, GA, USA). First, 0.2 g of each sample was out-gassed under a vacuum at 200 °C for 6 h prior to the measurement. Powder X-ray diffraction (PXRD) patterns were obtained from a PANalytical X'Pert PRO powder (Malvern, UK) diffractometer using Cu K$\alpha$ radiation ($\lambda$ = 0.1541 nm). The working voltage was 40 kV and the working current was 40 mA. The patterns were collected with a $2\theta$ range from 10° to 80° at a step of 0.0167°. The average crystallite sizes of the samples were calculated with the Scherrer equation based on the strongest hkl (111) diffraction peak of Co. Hydrogen temperature-programmed reduction ($H_2$-TPR) tests were accomplished on a GC-1690 chromatograph (Xi'an, China) with a TCD. Each catalyst (30 mg) was pretreated in $N_2$ (30 mL/min) at 200 °C for 30 min and cooled down to 50 °C. Then, the gas flow was switched to 5%$H_2$/Ar and the temperature was raised to 500 °C at the rate of 10 °C/min.

XPS tests were performed on a Thermo Scientific K-Alpha system (Waltham, MA, USA)) with Al K$\alpha$ radiation (1486.6 eV), operating at 150 W and with an energy pass of 20 eV. The spectra were calibrated with the carbon deposit C 1 s peak at 284.8 eV. Scanning electron microscopy (SEM) measurements were performed on a Nova NanoSEM 450 microscope (Eindhoven, The Netherlands). The samples were deposited onto a wafer attached to the sample holder with carbon tape, and then were sputter-coated with conductive gold to enhance the electrical conductivity before measurement. The TEM micrograph and particle size distribution of the catalysts were obtained on a TECNAI G2 20 instrument (Hillsboro, OR, USA) operating at 200 kV. The nanoparticle size distribution for each sample was determined using samples of ~200 nanoparticles. The EDS analysis was used to record elemental maps to determine the chemical composition. High-angle annular dark field (HAADF) images and elemental analysis (mapping) were obtained under STEM mode. The samples for the TEM study were previously dispersed in ethanol and deposited onto a perforated carbon film supported on a copper grid.

### 3.4. Computational and Modeling Methodology

Spin-polarized density functional theory (DFT) calculations were performed using the VASP code. The core electrons were described by the projector-augmented-wave method. The electronic exchange and correlation were described by a generalized gradient approximation method using the Perdew–Burke–Ernzerhof functional. The valence electrons were described by the Kohn–Sham wave functions being expanded on a plane-wave basis with the energy cutoff of 400 eV. The Brillouin zone was sampled with (3 × 3 × 1) Monkhorst–Pack k-point mesh. For convergence, all the configurations were defined as optimized when the forces of each atom fell below 0.05 eV/Å. The Co surface was modeled as a

4-layer, $4 \times 4(111)$ surface, with the lattice constant of 3.53 Å. The two top layers were relaxed without restrictions during structural optimizations, while the bottom two layers were fixed in bulk position, while the rest of the layers were allowed to relax.

To model different atomistic environments of Mn, Mn-on-Co(111), Mn-in-Co(111), and Mn-under-Co(111) were modeled, which, respectively, represent the Mn inside and outside of the Co lattice. The Mn-in-Co(111) was modeled by replacing a surface Co with Mn, while the Mn-on-Co(111) was modeled by placing a Mn atom on a Co(111) surface. The H, C, and CO binding energies were calculated, respectively, using the total energies of H, C, and CO (in vacuum) as the references.

## 4. Conclusions

High spatial proximity of the promoter and reactive metal is desired to maximize the effectiveness of the promoter. In this work, the as-synthesized bimetallic MOFs (CoMn-BTC) as self-sacrificial templates were thermally treated to fabricate FTS catalysts. The obtained CoMn@C catalysts inherited the original morphology of the precursor and possessed very high cobalt loading (52.7–33.0 wt.%), while containing well-dispersed active metal Co nanoparticles (13.1–13.4 nm) embedded in the carbon matrix, implying the optimal combination of high dispersion with maximum loading. Moreover, Mn preferentially formed isolated tiny particles rather than being embedded in the Co lattice or forming mixed Co–Mn oxides, and thus the highly dispersed MnO could yield plenty of Mn–Co interfaces. Although the close contact of Mn and Co on the surface with the bulk $Co^0$ remains stable, this preparation method seems to limit the formation of Co–Mn bulk solid solutions, even with Mn content close to 15.3 wt.% (1Co1Mn@C).

The incorporation of Mn significantly inhibits the production of $CH_4$ and $C_2$–$C_4$ paraffins, accompanying an enhancement in the light olefin ($C_{2\text{-}4}^=$) and $C_5^+$ selectivity. Accordingly, the methane selectivity decreases from 50.1% to 13.6%, and the selectivity towards long $C_5^+$ ranges from 23.5% over Co@C to 61.9% over 2Co1Mn@C. Furthermore, the intrinsic activity observed for the Mn-promoted catalysts increases with increasing Mn loading and peaks for the 2Co1Mn@C sample (CTY = $1.60 \times 10^{-5}$ $mol_{CO} \cdot g_{Co}^{-1} \cdot s^{-1}$) due to the abundant Co–Mn interfaces. The bimetallic MOF (CoMn-BTC) pyrolysis allows us to quantitatively place the Mn promoter in close association with Co, resulting in the weakened adsorption of H but enhanced adsorption strength of CO and C. Hence, the experimentally observed decrease in selectivity to $CH_4$ and the increased selectivity to light olefin and $C_5^+$ products when increasing the Mn/Co ratio to 0.5 are attributed to a decrease in the ratio of adsorbed H to CO on the surfaces of the Co nanoparticles. This work could pave the way for the potential use of bimetallic MOFs as catalyst precursors for the synthesis of promoted catalysts where intimate contact between the promoter and reactive metal is highly desirable.

**Supplementary Materials:** The following supporting information can be downloaded at: https://www.mdpi.com/article/10.3390/catal13030633/s1, Table S1. Surface elemental composition and atomic ratio measured by XPS, and Mn/Co molar ratio calculated by ICP-MS. Table S2. Surface area, crystallite sizes calculated by Scherrer equation, and TOF of MOF-derived catalysts with different Mn/Co molar ratios. Figure S1. CO conversion (A), $CH_4$ (B), $C_{2\text{-}4}^=$ (C), and $C_5^+$ selectivity (D) with time on stream (TOS) of the MOF-derived and active-carbon-supported catalysts. Here, the GHSV = 5000 mL $g_{cat}^{-1} \cdot h^{-1}$ for Co@C, GHSV = 8000 mL $g_{cat}^{-1} \cdot h^{-1}$ for 20Co1Mn@C, 10Co1Mn@C and 2Co1Mn@C, GHSV = 2400 mL $g_{cat}^{-1} \cdot h^{-1}$ for 1Co1Mn@C, GHSV = 4500 mL $g_{cat}^{-1} \cdot h^{-1}$ for 2Co1Mn/AC. (235 °C, 1 bar, and $H_2$/CO = 2). Figure S2. The SEM images for Co-BTC and 2Co1Mn-BTC. Figure S3. The XRD patterns for 2Co1Mn@C and reduced 2Co1Mn@C (reduction condition: 350 °C for 6 h under 10%$H_2$/Ar). Figure S4. The $H_2$-TPR profiles for MOF-derived catalysts. Figure S5. Calculated electron distribution of Mn-in-Co(111), side view (top) and top view (bottom). Blue and purple spheres represent Co and Mn, respectively. The cyan and yellow colors represent the depletion and accumulation of charge.

**Author Contributions:** Conceptualization, investigation, formal analysis, and writing—original draft preparation, L.Y.; methodology, Y.G. (Yu Gao); validation, Y.G. (Yupeng Guo); data curation, visualization and funding acquisition, Z.L.; writing-review and editing, J.C.; su-pervision and project administration, N.Y. and X.L. All authors have read and agreed to the published version of the manuscript.

**Funding:** This research was funded by the National Natural Science Foundation of China, grant number 22108253.

**Data Availability Statement:** The additional data are available in the separate Supplementary Materials.

**Acknowledgments:** The authors thank Zhiqi Liu and Wendi Chen for the DFT calculations.

**Conflicts of Interest:** The authors declare no conflict of interest.

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
