# Peer review of "Bimetal–Organic Framework-Derived CoMn@C Catalysts for Fischer–Tropsch Synthesis"

_catalysts, doi:10.3390/catal13030633_

Round 1
Reviewer 1 Report
It is an interesting paper about the synthesis of CoMn@C catalysts by carbonization of bimetal-organic frameworks with using different Mn/Co molar ratio. The obtained composites were examined in Fisher-Tropsch synthesis and fully characterized by nitrogen adsorption/desorption, XRD, XPS, FT-IR, SEM, TEM, HRTEM, HAADF-STEM and EDS. The theoretical calculation with DFT method was also performed to investigate activity and products selectivity of FTS performed with using synthesized catalysts. It is a meaningful work. Nevertheless, I suggest making some major corrections before its publication in Catalysts:
1. The isotherms presented in Figure 3 are not described in the manuscript. Which type of isotherm are they according to IUPAC classification? Why the hysteresis loop of 2Co1Mn@C isotherm is not closed? I suppose that this sample contains some impurities that left the sample during the degas process. Please reconsider the increase of temperature of sample drying or prolongation the time of this proces.
2. Why the PSD of 2Co1Mn@C is more narrow than that observed for Co@C? Is it caused by Mn addition?
3. Why did the authors not add to the manuscript the crystallite sizes of CaO and Co0 calculated by Sherrer aquation? Does the Mn addition influence on the crystallite sizes?
4. The Authors described that XP spectra did not show the shift of the Co0 binding energy. What about the peaks ascribed to the binding energies of Co2+, Co3+ and Mn2+? It seems that these peaks changed their position depending on Mn loading in catalysts. Consider the addition of this statement to the manuscript.
5. The conditions of FT-IR should be described in Materials and Methods section.
6. The same units of temperature should be used throughout the manuscript.
7. The title of manuscript is different in article and SD.
8. Lines 124 and 129: Figures 1a and 1b should be replaced by 2a and 2b.
Reviewer 2 Report
The paper describes CoMn catalysts for the FTS, prepared via bimetallic MOF structures that are subsequently pyrolyzed to make promoted cobalt catalysts.
The concept is interesting, but I find the paper somewhat unfinished. The Fischer-Tropsch synthesis is an experimentally demanding reaction to study, and in my opinion this paper needs to contain more information before it is published, especially on experimental detail and results regarding the activity measurements, but also regarding other aspects of the work.
Some detailed points.
1. The key property is the FTS activity. The preparation method would - as far as I can understand -give a material where the metal particles are encapsulated in carbon.
a. In the authors opinion: Is this the case?
b. If not, why don’t they oxidize when exposed to air?
c. Would the pre-reduction remove carbon from the samples, was this investigated (e.g using Temperature Programmed Hydrogenation)?
d. In my opinion characterization of the available cobalt surface area is needed (e.g. by hydrogen chemisorption), that would give the proper basis to compare activities.
e. The comparison should include a traditional catalyst, e.g. a traditional, supported cobalt catalyst tested at similar conditions. Only then can the claim of superior activity be made.
2. FTS experiments.
a. What is the particle size used for the catalyst material?
b. After what time was the activity recorded, and was there any development with time of the activity and selectivity (activation or induction period/deactivation, change in the selectivity with time)?
c. Was CO2 detected, and if so, was the selectivity dependent on the composition?
3. The materials appear to be microporous.
a. How can the activity and selectivity be explained without meso- or microporosity in the material? Or is there in deed macropososity, the SEM images indicate that there is?
b. Or are the particles so small that diffusion limitations are not an issue? (see also Q.2.)
c. Why are not all the samples characterized in terms of surface area, this should be included.
4. I’m not convinced by the interpretation of the XRD of the pyrolyzed samples (Fig. 3c). Could not this pattern also be explained by contributions from CO3O4? This is also indicated in the XPS; where both Co2+ and Co3+ are indicated.
5. The mean particle size was calculated to be 13 nm for all the samples. Is that the number average value (first moment of the distribution)? The distributions are not the same although the average size is – could that explain the differences in activity (estimates of the available metal surface area based on the second moment of the distribution)?
6. Minor QC issues
a. Ref 40 is missing from the list
b. Is the reaction T 230 °C (line 348) or 235 °C (line 98 and 358)?
c. I presume the XPS spectra were calibrated using the C1S peak at 284.8 eV (line 230), and not a peak at 248.8 eV (line 379)
d. Language: The manuscript would benefit from some language editing and further corrections, e.g. TEM (not TME) micrographs (383)
But, all in all, an interesting paper. Minor revision.
Reviewer 3 Report
Please see the attachment.

Round 2
Reviewer 1 Report
The authors corrected the manuscript according to the suggestions of the reviewers, which improved it. Thus, I recommend this work for publication in the Catalysts journal.
Author Response
We greatly appreciate you for the time spent making the useful suggestions.
Reviewer 3 Report
In the revised version, the authors have reasonably addressed most of the question raised by this reviewer to the original submission. However, there are still some points (see comments below) that require revision by the authors, especially regarding the calculation of TOF values reported in Table S2.
1. The XRD patterns of catalyst 2Co1Mn@C before and after reduction in H2, shown in Figure 2 of the author’s reply to comment #2 of my previous review, should be included in the Supporting Information and commented in the main text to support the statement that Co crystallite sizes are not significantly altered after the H2 reduction step.
2. Similarly, the H2-TPR profiles reported in Figure 3 of the author’s reply to my comment about the extent of Co reduction upon reduction of catalysts in H2 at 350 °C should be included in the Supporting Information and commented in the main manuscript.
3. The authors have to explain in detail how the TOFs given in Table S2 of the Supporting Information were determined, and which Co particle sizes (those derived from TEM or from XRD) were considered in the calculations. Moreover, the obtained values need to be carefully checked as, according to my own calculations, the TOFs reported in Table S2 are incorrect.
4. The two references provided in the answer to my comment #9 about the simplification of using metallic Mn instead of MnO in the DFT simulations have to be included in the manuscript to support the assumption made in the DFT calculations.
5. The English usage still needs further improvement. For instance, in P6L228, “more narrower” should be replaced by simply “narrower”.
